# Remote Sensing Evaluation Drone Herbicide Application Effectiveness for Controlling *Echinochloa* spp. in Rice Crop in Valencia (Spain)

**DOI:** 10.3390/s24030804

**Published:** 2024-01-25

**Authors:** Alberto San Bautista, Daniel Tarrazó-Serrano, Antonio Uris, Marta Blesa, Vicente Estruch-Guitart, Sergio Castiñeira-Ibáñez, Constanza Rubio

**Affiliations:** 1Departamento de Producción Vegetal, Universitat Politècnica de València, 46022 Valencia, Spain; asanbau@prv.upv.es; 2Centro de Tecnologías Físicas, Universitat Politècnica de València, 46022 Valencia, Spain; dtarrazo@fis.upv.es (D.T.-S.); sercasib@fis.upv.es (S.C.-I.); crubiom@fis.upv.es (C.R.); 3Escuela Técnica Superior de Ingeniería Agronómica y del Medio Natural, Universitat Politècnica de València, 46022 Valencia, Spain; marblevi@etsiamn.upv.es; 4Departamento de Economía y Ciencias Sociales, Universitat Politècnica de València, 46022 Valencia, Spain; vestruch@esp.upv.es

**Keywords:** remote sensing, Sentinel-2, agronomy, biomass, weeds, vegetation index, *Oryza sativa*

## Abstract

Rice (*Oryza sativa* L.) is a staple cereal in the diet of more than half of the world’s population. Within the European Union, Spain is a leader in rice production due to its climate and tradition, accounting for 26% of total EU production in 2020. The Valencian rice area covers around 15,000 hectares and is strongly influenced by biotic and abiotic factors. An important biotic factor affecting rice production is weeds, which compete with rice for sunlight, water and nutrients. The dominant weed in Spain is *Echinochloa* spp., although wild rice is becoming increasingly important. Rice cultivation in Valencia takes place in the area of L’Albufera de Valencia, which is a natural park, i.e., a special protection area. In this natural area, the use of phytosanitary products is limited, so it is necessary to use the minimum amount possible. Therefore, the objective of this work is to evaluate the possibility of using remote sensing effectively to determine the effectiveness of the application of the herbicide cyhalofop-butyl by drone for the control of *Echinochloa* spp. in rice crops in Valencia. The results will be compared with those obtained by using sterilisation machines (electric backpack sprayers) to apply the herbicide. To evaluate the effectiveness of the application, the reflectance obtained by the satellite sensors in the red and near infrared (NIR) wavelengths, as well as the normalised difference vegetation index (NDVI), were used. The remote sensing results were analysed and complemented by the number of rice plants and weeds per area, plant dry weight, leaf area, BBCH phenological state, SPAD index values, chlorophyll content and relative growth rate. Remote sensing is validated as an effective tool for determining the efficacy of an herbicide in controlling weeds applied by both the drone and the electric backpack sprayer. The weeds slowed down their development after the treatment. Depending on the phenological state of the crop and the active ingredient of the herbicide, these results are applicable to other areas with different climatic and environmental conditions.

## 1. Introduction

Rice (*Oryza sativa* L.) is a staple cereal that enters the diet of more than half of the world’s population [1]. Within the European Union, Spain, due to its climate and tradition, is a pioneer in rice production, accounting for 26% of total EU production, equivalent to 739 thousand tons in 2020 [2]. Valencia represents 15% of the Spanish production of rice, 112 million tons, spread over approximately 15 thousand hectares [3]. The rice crop area of Valencia is greatly affected by the influence of biotic and abiotic factors. Abiotic factors include salinity caused by the absence of crop rotation, low temperatures and droughts. In terms of biotic factors, the incidence of pests, such as stem borers like *Chilo suppressalis*, or diseases, such as rice blast, caused by the fungus *Pyricularia oryzae*, predominates. The amount of damage caused by pests and diseases can harm up to 25% of the annual rice production. However, another relevant biotic factor affecting rice production is weeds. The issue of weed infestation is the primary cause of significant yield loss globally [4]. Weeds are unwanted plants that grow in soil and compete with crops for essential resources like water, nutrients, sunlight, space and carbon dioxide. According to studies conducted by Buchanan [5], Cousens [6], Barbás et al. [7], Hasan et al. [8] and Mahé et al. [9], weeds can have a negative impact on crop growth and yield. Aside from causing direct damage, weeds also serve as a refuge for pests and diseases that can harm the crop [10], particularly *Pyricularia oryzae* in rice. Managing weeds is one of the biggest challenges in crop production [11].

The dominant weed in Spain is *Echinochloa* spp., although wild rice is also gaining importance [12]. Authors such as Brookes and Barfoot [13] quantified that the annual damage caused by biotic factors reduces production by up to 52%, of which about 21% is caused by pests. Weed infestation has a significant impact on the productivity of rice crops, as reported by Dass et al. [14], varying between 10 and 100% [4,15]. Dealing with weeds in rice is becoming increasingly complicated due to the emergence of herbicide resistance [16]. In addition, the number of active substances authorised in Spain and the EU is decreasing every year [17]. Globally, the traditional way of controlling weeds, pests and diseases in permanently waterlogged rice fields is complicated because the soil is not firm. The equipment traditionally used has been hand backpacks [18] or tractors with narrow rubber or metal wheels to which a hydraulic sprayer is attached [16]. Such tractors are powerful but heavy, which directly affects the soil surface through the formation of lanes [16]. In addition to these mechanisms, aircraft have also been used, jointly spraying large tracts of land [19], but this herbicide application is not allowed.

Rice cultivation in Valencia takes place in the area of L’Albufera de Valencia, which is a natural park of significant environmental wealth and is included in the catalogue of wetlands of the Valencian Community [20]; it was declared a Place of Community Interest (LIC) and a Special Protection Area for Birds (ZEPA) [21]. In addition, it is included in the Ramsar list of Wetlands of International Importance [22]. In this natural area, rice is the only authorised crop, but it must comply with a series of limitations in the use of phytosanitary products, infrastructure expansion, nutrient application, etc. The management of rice crops in Valencia is dependent on the use of phytosanitary products to control algae, weeds or fungal diseases [23], but the use of these has decreased in recent years.

Hand weeding is an effective method for weed control, but it is labour-intensive and not cost-effective in agricultural systems with high energy demands. Traditional herbicide applications with spray machines have a high economic and environmental impact. Therefore, it is necessary to develop new weed control systems that are both economically beneficial and environmentally friendly. In recent years, drones, or unmanned aerial vehicles (UAVs), have started to be used as a new tool in the field of agriculture and have become a technology widely used in precision agriculture [24]. In Japan, the first agricultural spraying applications using drones were conducted, and about 40% of Japan’s rice paddies, or approximately 2.5 million acres per year, were sprayed with aerial vehicles at the beginning of this century [25]. A more precise and uniform distribution can be achieved by applying active substances using drones [26]. They present a number of advantages over the rest, including the lack of a need for a special landing area, better mobility, lower water consumption, etc. [19]. In the case of their use in the application of phytosanitary products, previous studies conclude that herbicide applications with drones can replace applications with manual spray backpacks since similar leaf coverage is given [18]. Similar results were obtained by Wang et al. [27], who concluded that the amount of active matter deposited on wheat did not vary between drone and conventional backpack applications. In another study on aphid control in wheat with a drone bar sprayer and conventional backpack sprayer, Wang et al. [27] concluded that the use of drones for pest control could be a viable strategy. Not only can drone technology be used for pest control but also for weed control, and in particular, Pranaswi et al. [28] confirmed effective weed control in wheat with combined drone applications. Supriya et al. [29] reported satisfactory results in weed control in corn crops using pre- and post-emergence herbicides applied by drone. Ahmad et al. [30] reported that drone operating parameters, such as height and flight speed, can improve herbicide application. Zhang et al. [31], in their studies, concluded that the drone flight speed and height also have a significant effect on field airflow distribution.

Nowadays, it is possible to use remotely sensed hyperspectral imagery to classify weeds [32,33] and to monitor weed growth based on chlorophyll content and plant cover [34], making it possible to monitor herbicide applications using both drones and sterilizing machines. Therefore, the objective of this work is to study the effectiveness of remote sensing to verify the possible improvement in the application with drones of a herbicide with the active ingredient cyhalofop-butyl compared to its application with sterilizing machines for the control of *Echinochloa* spp. in rice crops in Valencia, maintaining yield and minimizing rice toxicity by the herbicide. As it is a natural park with special protection, the use of phytosanitary products is limited and highly controlled. Therefore, the application of phytosanitary products must be as effective as possible to minimise phytosanitary residues in the park.

## 2. Materials and Methods

### 2.1. Design of the Experiment

The field experiment was carried out between May and June 2021 to evaluate the effect of the application of the cyhalofop-butyl herbicide by different methods on the rice crop. The experimental plot was located in the municipality of Sollana within the Albufera Natural Park, province of Valencia, Spain (latitude: 39°19′21.6″ N, longitude: 0°24′3.9″ W) (Google Earth, 2021). The experimental plot is a field owned by a farmer from the neighbouring municipality. The chosen field corresponds to polygon 8, plot 469, with an approximate area of 1.4 hectares. The plant material tested was rice of the “Sirio” variety, which is a clearfield variety that is resistant to the application of certain herbicides such as cyhalofop-butyl, which allows good weed control. Rice sowing was carried out on 28 May 2021, 4 days after the plot was flooded. The application of the herbicide at the beginning of the experiment was carried out on 4 June 2021, 7 days after sowing (das).

The study herbicide is cyhalofop-butyl as a post-emergence treatment. It is a selective herbicide for the control of certain grasses considered weeds in rice. The weeds to be controlled are *Echinochloa* spp., *Leptochloa* spp., *Panicum dicothomiflorum*, *Paspalum distichum*, *Digitaria sanguinalis* and *Setaria* spp. Against *Echinochloa* spp. plants, which is the main weed of the crop, it is considered effective if it has between 1 and 4 leaves; for this reason, the time of application must be correctly chosen. The formulation of the herbicide used in the experiment is an emulsifiable concentrate (EC), which incorporates a wetting agent for better contact with the plant. The active ingredients are cyhalofop-butyl (200 g/L) + Polyglycol 26-2N (333.3 g/L) [35].

Post-emergence cyhalofop-butyl herbicide is characterised as a systemic herbicide of the aryloxyphenoxy-propionate family which acts through the inhibition of the coenzyme acetyl carboxylase (ACCase). It is absorbed by leaves and stems and controls plants of the genus *Echinochloa* spp. [36].

The recommended application rate of the cyhalofop-butyl product is 1.5 L ha^−1^. The drone applied 39.93 L ha^−1^ (2.3 L in each subplot), maintaining the recommended herbicide dose, and the drone tank (6 L) was prepared with 225 mL of cyhalofop-butyl and 5775 mL of water to achieve the correct concentration and dose. In the case of the electric backpack sprayer, 173 L ha^−1^ was applied, and in each subplot, 9.97 L of fluid was needed. To avoid air aspiration, the tank was always filled with the 15 L allowed. For each 15 L tank, 130 mL of cyhalofop-butyl and 14,870 mL of water were added, maintaining the recommended application rate.

The experiment consisted of three treatments: drone (DR), spray machine (SM) and control (C). It was designed in randomised blocks with three replications in subplots of 576 m^2^ (24 × 24 m), which resulted in three blocks with three subplots each, corresponding to the different treatments. During application, the plot presented uniformity in the water sheet (almost non-existent) and uniformity in the plant population. As for the conditions of the field experiment, it was carried out without the presence of a water sheet but at its field capacity and with uniformity in the plant population. Taberner [37] has already used the cyhalofop herbicide for the control of *Echinochloa* spp. in rice with water level at the time of the test (waterlogged soil, flooded, but without water cover), similar to the experiment of the study in this report. To avoid cross-contamination between plots, sampling was randomly arranged in the centre of the subplots for each of the data collection dates.

### 2.2. Application Equipment

To evaluate and compare the efficacy of the application of the cyhalofop-butyl herbicide (selective for *Echinochloa* spp.) with an UAV in the rice crop, a drone and an electric backpack sprayer were selected. The efficacy of the application was compared according to the number of rice and weed plants per area, the plant dry weight, the leaf area, the BBCH phenological stage, the SPAD index values, the chlorophyll content and the relative growth rate.

The aerial spraying equipment used was a drone manufactured by AGR (Zhejiang, China), which is a supplier of high-end aerial equipment for plant protection, which gives its name to its commercial brand. The model used was a four-rotor A6 phytosanitary protection drone. The dimensions with closed blades are 855 × 855 × 409 mm and with open blades, 970 × 970 × 300 mm. The unloaded weight of the equipment is 7.1 kg with a maximum load of 15.6 kg. The volumetric capacity of the equipment is 6 L, and the pump guarantees a flow rate between 1 and 1.8 L min^−1^, with a pressure of up to 5 bar and a power of 12 V. At the end of each arm, under each of the rotors, each of the 4 nozzles that compose it is located. It uses 24 V batteries for its operation (AGR-A6, n.d.).

Regarding the ground spraying equipment, an electric hydraulic sprayer was used for the application of phytosanitary products, manufactured by Grupo Sanz (Sanz Hermanos S.L., Benisanó, Valencia, Spain) (Pulmic Pegasus 15 Electric Sprayer—Grupo Sanz, n.d.). This sprayer is attached to a carbon fibre bar of its own experimental design and construction, with a working width of 4 metres and 8 nozzle holders separated by 50 cm. This sprayer incorporates a membrane pump with three application speeds with a range of 1–3 bar. It is powered by a lithium battery with an autonomy of up to 7 h, a maximum flow rate of 2.2 L min^−1^ and a tank capacity of 15 L.

Both devices were calibrated prior to use to ensure their correct operation and to be able to design the application accurately. For the calibration of the aerial equipment, several nozzles were tested: Li-Cheng KZ80-0.6 green hollow cone nozzles (Ningbo Licheng Agricultural Spray Technology Co., Ltd. Jinsheng, China); Teejet XR 110 01 VH orange (Fede Pulverizadores, Cheste, Valencia, Spain); Agroplast Ppij 6MS 01C orange anti-drift with ceramic tip (Agroplast gmbh, Rabenau, Germany); and Albuz TVI 80 pink anti-drift (Albuz, Évreux, France). The nozzles selected for the application were Agroplast Ppij brand, model 6MS 01C, orange, anti-drift with a ceramic tip, fan-shaped, 110°, manufactured by Agroplast gmbh, Rabenau, Germany (Agroplast Sistemas de Riego—Manufacturer of Technical Products for Drip Irrigation, n.d.). These are the most suitable nozzles as the pump operates at 6 bar, the maximum capacity provided is measured at the nozzle, the flow rate per nozzle is 0.57 L min^−1^ and a correct fan opening is achieved at a height of 1.5 m, achieving a working width of 2 m and an application volume of 39.93 L ha^−1^.

In the case of the ground equipment, the pressure at the bar end is set at 1.7 bar with a manometer reading of 1.8 bar. For the nozzles, ASJspray-Jet Hypro model F-110 015 green fan nozzles, manufactured at ASJspray-Jet, Italy, were used (ASJ NOZZLE Spray Nozzles, n.d.). The nominal flow rate of these nozzles at 1.7 bar is 0.43 L min^−1^, and the average calibration readings were 0.42 L min^−1^ (maximum recorded flow rate, 0.435 L min^−1^, and minimum, 0.410 L min^−1^). Good fan formation was observed at the selected pressure. In addition, the forward speed was determined in the field and was 0.806 m s^−1^ (2.9 km h^−1^). The flow rate of the equipment was 3.36 L min^−1^, and with the calibrated data set, the application volume was 173 L ha^−1^.

### 2.3. Monitoring Using Sentinel-2 Images

One of the key technologies that has improved agriculture in recent years is remote sensing. Satellite sensors obtain vegetation reflectance data at different wavelengths. In addition to obtaining multispectral data, they are able to obtain the time series of multitemporal reflectance data. Depending on the phenological state of the vegetation, the reflectance of the vegetation is high in the green wavelength region and highest in the NIR region, while it is low in the blue and red wavelength regions of the visible spectrum. Thus, by algebraically combining these bands, in addition to the vegetation indices, different vegetation characteristics can be obtained.

Sentinel-2 satellite images obtained with a Multi-Spectral Instrument (MSI) on board the Sentinel-2A/B constellation were selected from the subplots of the experiment. Sentinel-2 captures images of the Earth’s surface in 13 spectral bands with a temporal resolution of 5 days, of which those corresponding to a 10 m spatial resolution are used for the study. The T30SYJ tile was used, selecting cloud-free dates from 9 June to 24 June 2021, which were 9, 14 and 24 June. The downloaded product was 2A, with an atmospheric correction carried out by ESA, and the images were obtained from the Copernicus website [38]. Table 1 shows the main characteristics of Sentinel-2 and the most relevant bands (Bi) analysed in the study: red (B4) and NIR (B8). The data obtained for each band and the B8/B4 ratio were represented for the period studied.

From the values of the analysed bands, the normalised difference vegetation index (NDVI) calculated for each set of trial subplots with the same treatment is also obtained. The NDVI has a range of 0–1, with a proximity to 1 indicating greater plant height and greener shade. The NVDI corresponds to the quotient between the difference between (B8 − B4) and the sum of both (B8 + B4). Thus, these indices serve to maximise the sensitivity of the information provided by the satellite spectral bands to a biophysical parameter [39]. All the data (band, ratios and vegetative index) were normalised by feature scaling to bring the values into the range [0, 1].

In addition, a prediction of rice crop production was made for each of the subplots using the model proposed by Franch et al. [40]. The satellite images were processed with the QGIS 3.10.14 software.

### 2.4. Plant Measurements

For the validation of the drone as a suitable system for the application of the active material cyhalofop-butyl in the control of *Echinochloa* spp. plants in the rice crop, both its efficacy and the absence of phytotoxicity in the rice crop were evaluated.

The weed density was calculated by placing a 0.25 m^2^ quadrant at four random spots in each plot and repeating it three times. Agronomic parameters and morphological and physiological indicators were evaluated every 7 days, following the technical prescription of herbicide, two times. The first evaluation was carried out 14 days after sowing (das_14_), and the second 21 days after sowing (das_21_). For each data collection, the number of *Echinochloa* spp. and rice plants per m^2^ was recorded using a 50 × 50 cm structure. The SPAD (Soil Plant Analysis Development) index, a relative measure of the level of chlorophyll in leaves with a range of 0–199, was also measured with a chlorophyll meter MC-100 (Apogee Instruments Inc., Logan, UT, USA). The dry weight of *Echinochloa* spp. and rice plants per m^2^ was measured after they were oven-dried at 105 °C to a constant weight. The leaf area of the rice plants was measured using ImageJ software (version 1.53e) based on an image analysis. The phenological status of both the *Echinochloa* spp. and rice plants was determined according to the BBCH scale.

The total chlorophyll, a and b, and carotenoid (μmol m^−2^) contents of the rice plants were measured using the coefficients for rice described by Parry et al. [41] and by extraction with acetone, according to the protocol established by Porra et al. [42]. The absorbance (ABS) of the samples was quantified using a spectrophotometer at three wavelengths (662 nm, 645 nm and 470 nm), according to the following equations:(1)$$\mathrm{Chlorophyll}\;a=12.25\;\times \;\mathrm{ABS}662\;-\;2.55\;\times \;\mathrm{ABS}645\;(\mu g\;{\mathrm{mL}}^{-1}),$$
(2)$$\mathrm{Chlorophyll}\;b=20.31\;\times \;\mathrm{ABS}645\;-\;2.55\;\times \;\mathrm{ABS}662\;(\mu g\;{\mathrm{mL}}^{-1}),$$
(3)$$\mathrm{Carotenoids}:\;((1000\;\times \;\mathrm{ABS}470)\;-\;(1.82\;\times \;(\mathrm{Chlorophyll}\;a))\;-(85.02\;\times \;(\mathrm{Chlorophyll}\;b))/198\;(\mu g\;{\mathrm{mL}}^{-1})$$

In addition, with the previously obtained results of the dry weight (DW) of both the *Echinochloa* spp. and rice plants, the relative growth rate (RGR) was calculated, defined as the variation in the biomass per unit of time, for which two results from two dates are used [43,44]. The expression to be used for the calculation of the relative growth rate is the following, according to two times after sowing (DW_14_ and DW_21_)
(4)$$\mathrm{RGR}\;\left(\%\right)=\frac{\ln({DW}_{21})\;-\ln({DW}_{14})\;}{{\mathrm{das}}_{21}-{\mathrm{das}}_{14}}$$

The influence of the parameters studied on rice plants and weeds was assessed by means of an analysis of variance (ANOVA). The mean separations were measured when appropriate using the least significant difference at *p* < 0.05. The statistical program used was Statgraphics Centurion XVIII.

A flow chart of the study is shown in Figure 1.

## 3. Results

### 3.1. Remote Sensing Data

To evaluate the effectiveness of the application, both with drones and spray machines, of the active material cyhalofop-butyl in the control of *Echinochloa* spp. plants in rice cultivation, remote sensing was used. Figure 2a,b show the values of the Sentinel-2 satellite electromagnetic bands (B4 and B8) of the treatments for the available cloud-free dates (9, 14 and 24 June).

During the first two moments of recording reflectance values in B4 and B8 (9 and 14 June), no significant differences were found between the different herbicide application systems and the control. However, a slight change in the reflectance of the spectral bands was observed in the control treatment between 9 and 14 June, with a decrease in the B4 band and an increase in the B8 band. This trend was confirmed by the record made on June 24. At this time, statistically significant differences were observed between the two application methods (DR and SM) and the control. A lower reflectance value was observed in B4 in the control plots (C), with statistically significant differences (*p* < 0.05) compared to the DR and SM plots and no statistically significant differences between DR and SM.

The results of the NDVI (vegetative index) are shown in Figure 2c. On 9 and 14 June, the NDVI values were relatively low because not all the soil was yet covered by plant material and soil and water were still visible, and no statistically significant differences were found between the treatments. It was on 24 June when the total area of the subplots was occupied and covered by plant material, and it was possible to distinguish between possible unwanted weeds and the rice crop itself. At that time, the NDVI values were higher in the plots treated with DR and SM, with statistically significant differences with respect to the control plot (*p* < 0.05). Similar results were obtained with the B8/B4 ratio at the end of the study period (24 June), as shown in Figure 2d. The highest values of this ratio showed statistically significant differences with respect to the plots treated with DR and SM (*p* < 0.05), evidencing a possible vegetative cover of weeds in the plots not treated with the herbicide (control).

These results reinforce the possibility of using the drone as a complementary piece of equipment or as a replacement for the conventional ground treatment in the control of *Echinochloa* spp. 7 das with the cyhalofop-butyl herbicide. At the same time, it is confirmed that the treatment with cyhalofop-butyl did not cause differences in the vigour and vegetative growth of the rice plants 14 das.

Following the model proposed by Franch et al. [40], the capacity of the subplots is estimated, as shown in Figure 3. From the results obtained, it is verified that there are no significant differences between the three treatments. The application of the herbicide with the drone and spray machine did not damage the crop and, therefore, did not reduce the expected production. Apparently, the control treatment shows the lowest data, 7102.96 (kg ha^−1^), but this difference is not considered significant, so all the subplots will have a similar production capacity.

The use of the Sentinel-2 satellite and the performance estimation model defined by [40] allow for the estimation of rice crop production 90 days before harvesting. The absence of significant differences between the treatments also shows that the higher concentration of herbicide used in the drone application did not impose a toxic effect on the rice plants, allowing for the correct development of the rice with a similar performance to the rest of the treatments.

For the validation of the remote sensing results obtained through the application of the active material cyhalofop-butyl to control *Echinochloa* spp. plants in the rice crop, both its effectiveness and the absence of phytotoxicity in the rice crop were evaluated. For this, agronomic parameters and morphological and physiological indicators were used.

### 3.2. Biomass and Growth Indexes in Rice Plants

As productivity parameters, the number of plants per square metre, the dry weight, the relative growth rate and the leaf area were considered, reflecting the presence or absence of phytotoxicity in the rice crop. Table 2 shows the results of these parameters 14 days after sowing. Considering the dry matter content, relative growth rate and leaf area, there were no statistically significant differences for the study factor (treatment), and consequently, there were no significant differences at the treatment levels. The absence of statistically significant differences in the leaf area reflects the absence of phytotoxicity in the crop as well as the absence of differences in the RGR. The application of the cyhalofop-butyl herbicide at a high concentration with a drone does not cause any damage to the crop.

The evolution of the rice plant productivity parameters was not affected by the number of plants per surface area, as no differences were found between the control, drone and spray machine treatment groups. The small variations in plant numbers were due to the variability present in the field caused by sowing. These results show that the post-emergence herbicide application with the drone does not limit the growth of the rice plants and is comparable to that carried out by the spray machine (see Appendix A).

### 3.3. SPAD Index and Pigment Concentration in Rice Plants

The results concerning the concentration of pigments in the rice plants and the SPAD values are shown in Table 3. Regarding the SPAD readings for the rice plants 14 days after sowing, the three treatments showed no s.e. differences. In relation to the levels of chlorophylls a and b and the total chlorophylls and carotenoids, no statistically significant differences in the effect of the treatments were found, and consequently, there were no statistically significant differences for the presence of these pigments. The SPAD values, together with the levels of chlorophyll a and b and the total chlorophyll and carotenoids contents, showed the absence of phytotoxicity in the rice crop. The nine subplots followed parallel crop development. The analysis of the symptomatology in the field already indicated that no phytotoxicity problems were observed, and all the plots developed and evolved in a similar way, which shows the absence of a toxic effect of the drone application on the rice plants.

### 3.4. Growth Kinetics of Echinochloa spp.

To evaluate the efficacy of the application of the cyhalofop-butyl herbicide, the number of *Echinochloa* spp. plants per square metre was counted, and their growth kinetics were studied, as shown in Table 4. The results show the differences between the treatments. The differences are appreciated both through the number of plants and through their dry weight 14 and 21 days after sowing. In both aspects, the significance level of the Fisher-Sendecor statistical test for the attended factor is *p* < 0.05.

The largest number of *Echinochloa* spp. plants was obtained in the control plots, as shown in Table 4, with no statistically significant differences detected between the drone and spray machine treatments (*p* > 0.05) for this parameter 14 and 21 days after sowing. The dry weight gain of the *Echinochloa* spp. plants followed the same behaviour. Even with these differences, no statistically significant differences between the treatments were found for the relative growth rate (RGR) of the *Echinochloa* spp. plants.

Regarding field parameters indicating efficacy, both the number of *Echinochloa* spp. plants and their dry matter m^−2^ at 14 and 21 das (7 and 14 days after treatment) show differences between the treated and untreated subplots. The values are always higher in the control subplots. However, these same results were corroborated by the Sentinel-2 satellite’s electromagnetic band values, as was is able to find differences between the treatments.

### 3.5. Phenological Stage

The effectiveness of the applications in relation to the control of the unwanted *Echinochloa* spp. plants is reflected in the samples taken 14 and 21 days after sowing, as we detected notable differences in the phenological stage of the *Echinochloa* spp. plants between the control plants versus the plants that underwent the drone and spray machine treatments, as shown in Figure 4. In the control plots, the *Echinochloa* spp. plants had already started tillering (BBCH 21–22) and in general showed one or two suckers. In the drone- and spray-machine-treated plots, the plants showed no tillering but had three or four leaves open (BBCH 13–14) or one leaf initiating development. This was because the herbicide used was highly effective against the *Echinochloa* spp. plants with up to two leaves. The more developed plants weakened but continued their vegetative cycle, and 14 days after sowing, they showed 3–4 leaves, which showed slowed growth. The *Echinochloa* spp. plants found with only one leaf (BBCH 10–11) were those that, at the time of treatment, had not yet emerged from the surface.

As for the phenological stages visualised in the rice, these corresponded to what is expected for the crop at each time point, and like the leaf area, they did not reflect symptoms of phytotoxicity. Regarding the *Echinochloa* spp. plants, their phenological stages 14 days after sowing (7 days after treatment) already showed evidence of the treatment’s efficacy. In the subplots treated with DR or SM, there was no tillering of the weed plants, while in the control plots, one or two tillers were already showing.

## 4. Discussion

Based on the results obtained through remote sensing and the agronomic parameters and morphological and physiological indicators evaluated in the rice crops, it can be observed that the drone application of cyhalofop-butyl is effective for the control of *Echinochloa* spp. in its early stages.

The results of the cyhalofop-butyl herbicide application 7 days after rice planting for the *Echinochloa* spp. control can be monitored by Sentinel-2 images in the B4 and B8 electromagnetic reflectance bands during the first 27 days after sowing. This study is possible because the relationship between radiation in the red band and green biomass is an inverse nonlinear relationship, while in the near-infrared band, the nonlinear relationship is direct [46], resulting from the strong absorption of incident radiation by chlorophyll. In contrast, the near-infrared (NIR) spectral region does not absorb much solar radiation and does not correlate significantly with vegetation pigments [47]. Instead, it is highly correlated with the crop biomass and leaf area index (LAI) [48], showing greater reflectance in the NIR band with a higher biomass or LAI.

In this sense, in our experiment, from the reflectance values in these bands, it is possible to detect the vegetative growth of the weeds by the higher intensity in the green colour of their leaves due to the relationship between the B4 band and intensity of the green colour of the leaves of the plants, as opposed to the more chlorotic colour of the rice in the early stages of plant growth due to the effect of the herbicide. In addition, higher values of the B8 band were detected in the control plots, which indicates that the area covered per plant is much higher due to the presence of weeds in the crop because of the relationship between the B8 band and the canopy of the plants.

Not only is it possible to use hyperspectral imaging to discriminate between the spectral signatures of some weeds after herbicide applications [32], but it is also possible to study the dynamics of growth in rice plants and *Echinocloa* spp. from the evolution of reflectance in the red and NIR bands. So, monitoring rice plants and *Echinocloa* spp. could be effective in gaining a deeper understanding of crop–weed competition dynamics, which is necessary for effective integrated weed management [49,50].

The first vegetation index proposed was the NDVI, calculated as the difference between near-infrared radiation (NIR) and red radiation divided by the sum of both radiations [51]. The NDVI is a useful tool for characterizing plant growth and studying soil, as well as for determining how different types of stress affect crops. Using this vegetation index in this study, the behaviour of the rice crop in relation to the presence of weeds was confirmed by the evolution of the NDVI in the period studied (before 27 days). The vegetative index (NDVI), which is directly related to a greater formation of green biomass in an area of vegetative cover, was higher in the plots not treated with herbicide compared to the values recorded in the plots treated with DR and SM. The NDVI value saturated after 17 days in the untreated plots, a result that coincides with those obtained by other authors [52], as a consequence of the growth of weeds. On the other hand, in the plots treated with DR and SM, no saturation of the value was observed as a consequence of the effect of less growth in the rice plants compared to the higher density obtained by the weed plants.

The use of the Sentinel-2 satellite and the performance estimation model defined by Franch et al. [40] allow for the estimation of rice crop production 90 days before harvesting. The absence of s.e. differences between the effects of the treatments DR and SM on yield was also reported by Jeevan et al. [53] using less water volume per ha with DR. Also, these results show that the higher concentration of herbicide used in the drone application did not impose a toxic effect on the rice plants, allowing for the correct development of the rice and a similar performance to the rest of the treatments.

The yield estimation of the rice crop using the models developed by the research group confirms that the herbicide application with the drone is satisfactory for weed control. However, it is necessary to note a null effect on rice plants and a significant reduction in *Echinocloa* spp. plants. In other works, it has been confirmed that the drone application of cyhalofop-butyl is effective for the control of *Echinochloa* spp. According to Naveen et al. [54], using herbicides through drones has improved productive parameters such as panicle number and grain yield. Moreover, it has reduced the amount of dry biomass in weeds. However, the authors suggest using drones for the application of cyhalofop-butyl post-emergence herbicide at the recommended dose to obtain more effective control of weeds. But it should not be forgotten that Ortiz [55] showed a 14.6% resistance of *Echinochloa colona* (L.) accessions from different origins to the cyhalofop-butyl herbicide in rice, which serves as a wake-up call to encourage the use of herbicides with different modes of action, more focalised applications or even new mechanisms to control weeds, as well as monitoring the effectiveness of applications in the coming years. Planas [56] highlighted innovations in precision agriculture and crop protection, including the use of drones to control pests and diseases in low-growing and woody crops. He also talked about the incorporation of sensors for the application of doses proportional to the leaf area detected. These types of technologies have increased over the years and are some of the improvements that must be incorporated to improve efficiency and reduce costs for farmers.

Neither the number of rice plants nor the dry biomass of the rice plants were affected by the herbicide treatments (DR and SM). Similarly, Arockia et al. [57] reported lower weed density and lower dry matter after pre- and post-emergence herbicide applications in rice crops by drone. Other advantages could be attractive to study, such as a lower amount of water use and lower labour requirements, as demonstrated by Hiremtath et al. [58]. Studies carried out by Martin [18] on spray deposition on weeds from a remotely piloted aerial application system also confirm the results obtained in our study. Similar results were obtained for other cereal crops like wheat or maize with pre- and post-emergence herbicides applied with a drone [28,29], and it was concluded that one important decision was the herbicide dose.

Similarly, the growth indices used for the study of growth seasons (RGR and LAI) were not modified by the two application systems studied. These results show that the drone application is comparable to the SM application and that the higher herbicide concentration (not the dose) used in DR is not harmful to rice plants. As proof of this, no significant reduction in the most important plant pigments, chlorophyll a and b and carotenoids, was observed in either of the two herbicide application systems.

The phenological stage of the rice in the three treatments 14 days after sowing was 21 according to the BBCH scale, corresponding to the beginning of tillering [45]. In most of the plants, some tillering was already visible.

However, the effect of the herbicide application with DR and SM caused a significant reduction in the *Echinocloa* spp. plants, affecting the number of weed plants and the crop biomass, although the relative growth rate (RGR) was not found to be modified by the herbicide action on the *Echinocloa* spp. plants. The herbicide effect caused a reduction in the number of plants but did not have any influence on the relative growth rate of the plants not affected by the herbicide.

The results obtained are encouraging, and drones are interesting tools for use in weed control in rice, but certain parameters must be taken into account and established rigorously prior to their application. A more precise adjustment of the equipment and field implementation are pending tasks for future studies. Authors such as Mayanquer [59] studied the calibration and precise adjustment of a drone for carrying out phytosanitary treatments. It must be taken into account that each piece of equipment is different, and this action is necessary for the optimization of its proper functioning. Wang et al. [27] compared four types of UAV sprayers and found that height and flight speed had a pronounced impact on the performance of the vehicles. Flight speed had a pronounced impact on droplet penetrability. In their study, droplet penetrability was inversely proportional to flight speed and height; therefore, these parameters need to be set and fixed according to the crop and its time of development.

The monitoring of herbicide applications by a drone using Sentinel-2 images could help us study and better understand the dynamics of weed growth in rice crops for better integrated weed management. Furthermore, they could be very beneficial for weed control, as they are more sustainable in economic and environmental terms, as shown by other studies [57,60,61,62], especially in more protected natural environments such as the Albufera Natural Park (Valencia, Spain). Several studies have shown that herbicides can be used to control invasive plants in managed wetlands [63].

## 5. Conclusions

This work analyses the validity of using remote sensing to evaluate the effectiveness of the application of the herbicide cyhalofop-butyl at high doses in the early stages of rice growth (7 days after sowing). The results were compared when the herbicide was applied using a spraying machine and when it was applied using drones. According to the results obtained from remote sensing and the analysis of productivity parameters, the SPAD index and the concentration of pigments in the leaves of the rice crop, the absence of phytotoxicity in the crop was shown after the concentrated application of the herbicide cyhalofop-butyl, both with a drone and spraying machine. Similarly, the application with a traditional spraying machine does not generate phytotoxicity either. The growth kinetics of the *Echinochloa* spp. plants show efficacy in their elimination after the application of the herbicide with the drone, as with the spraying machine. The weeds slow down their development after treatment, and the phenological state of the weeds in each of the treatment subplots also corroborates the efficacy of the treatment. Finally, remote sensing electromagnetic bands and NDVIs are able to certify the efficacy of these treatments. Furthermore, they are able to validate the absence of a phytotoxic effect because they do not reflect differences between the treatments DR and SM and help estimate the final yield for each treatment 90 days before harvesting without non-significant differences.

The experiment reported the feasibility and high efficiency of using remote sensing to evaluate the effectiveness of herbicide applications in rice crops. Although the results obtained in this work are based on rice crops, they could be applied to other crops, depending on the phenological state of the crop and the active ingredient of the herbicide. These results could also be applied to other areas with different climatic and environmental conditions.

## Figures and Tables

**Figure 1 sensors-24-00804-f001:**
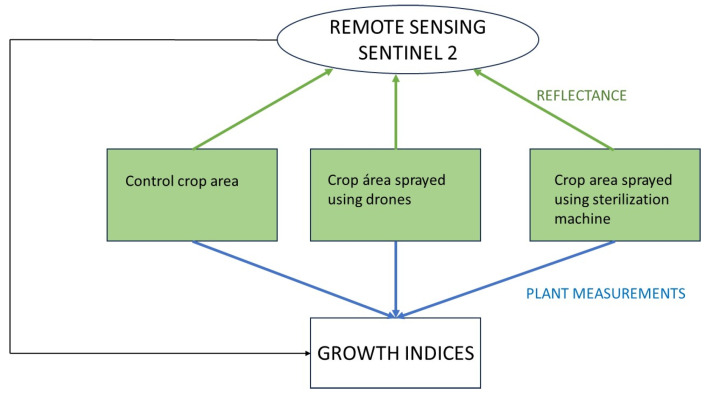
Flow chart of the study.

**Figure 2 sensors-24-00804-f002:**
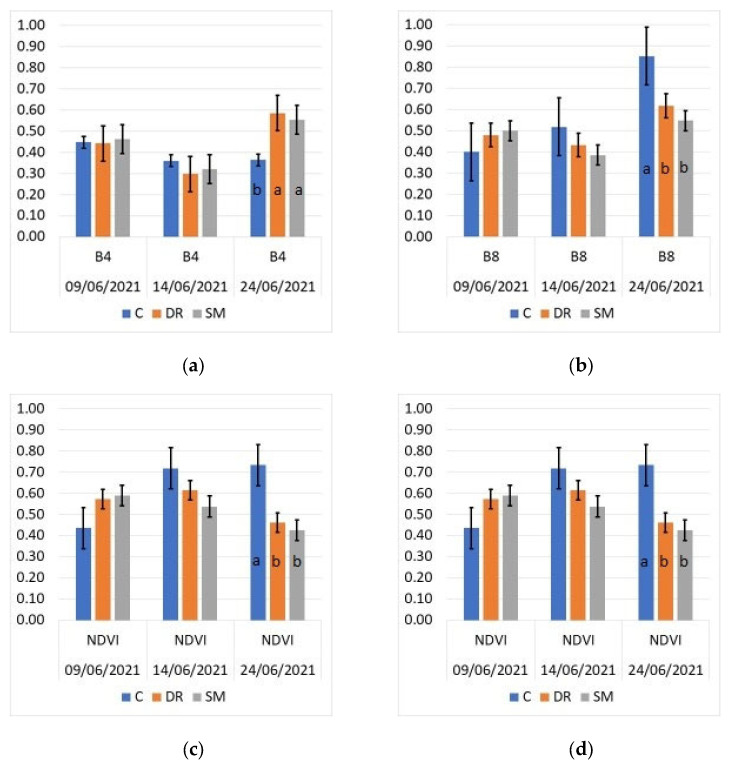
Values of the electromagnetic bands and indices obtained by the Sentinel-2 satellite on 9, 14 and 24 June (after treatment with cyhalofop-butyl). (**a**) B4, (**b**) B8, (**c**) NDVI and (**d**) B8/B4. Different letters indicate statistically significant differences using LSD test (*p* < 0.05). Vertical bars indicate standard errors.

**Figure 3 sensors-24-00804-f003:**
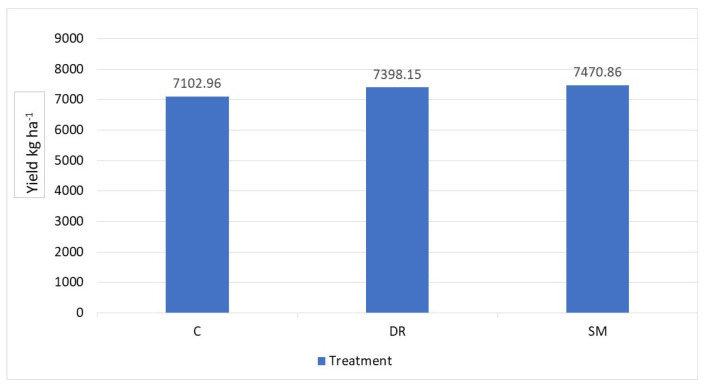
Marketable rice production yield estimate per treatment for each treatment group (C, DR, SM), according to [40].

**Figure 4 sensors-24-00804-f004:**
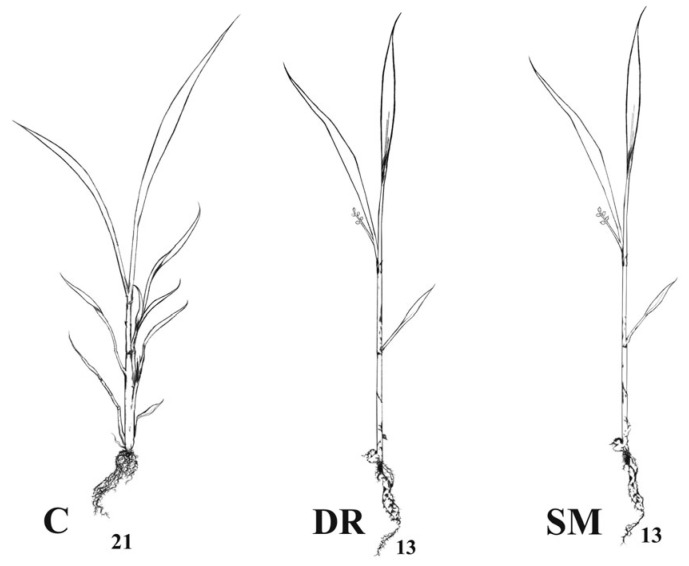
Phenological stage of *Echinochloa* spp. plants 14 days after sowing in treatments C, D and B. Illustrated by Cuca Nacher adapted from Enz M. and Dachler Ch., 1998 [45].

**Table 1 sensors-24-00804-t001:** Main characteristics of the Sentinel-2 spectral bands analysed.

Bands	Central Wavelength (nm)	Spatial Resolution (m)
B4-Red	665	10
B8-NIR	842	10

**Table 2 sensors-24-00804-t002:** Values of rice productivity parameters: number of plants per square meter; dry weight (g·m^−2^), relative growth rate (d^−1^) and leaf area (cm^−2^·pl^−1^) 14 days after sowing.

Herbicide Application	Number of Plants m^−2^	Dry Weight (g·m^−2^)	Relative Growth Rate (d^−1^)	Leaf Area (cm^−2^·pl^−1^)
Control (C)	608.00	21.31	0.2831	9.88
Drone (DR)	604.00	18.21	0.2962	8.30
Spray machine (SM)	659.67	19.04	0.2932	8.96
Probability	0.8461	0.8457	0.6663	0.4875

**Table 3 sensors-24-00804-t003:** SPAD, carotenoids and chlorophyll (a, b and total) values in rice plants 14 days after planting.

Herbicide Application	SPAD	Carotenoids (mg·g^−1^)	Chlorophyll		
			a (mg·g^−1^)	b (mg·g^−1^)	Total (mg·g^−1^)
Control (C)	11.48	0.071	0.238	0.120	0.343
Drone (DR)	11.00	0.074	0.226	0.116	0.357
Spray machine (SM)	11.41	0.075	0.236	0.120	0.360
Probability	0.8313	0.7813	0.8313	0.9130	0.8831

**Table 4 sensors-24-00804-t004:** Number of *Echinochloa* spp. plants per square metre and growth kinetics of *Echinochloa* spp. plants (number of plants per square metre, dry weight (g m^−2^) and relative growth rate (RGR) (d^−1^)) 14 and 21 days after sowing. Different letters indicate statistically significant differences using LSD test (*p* < 0.05).

Herbicide Application	Number of Plants m^−2^		Dry Weight (g·m^−2^)		Relative Growth Rate (d^−1^)	
	14 Das	21 Das	14 Das	21 Das	14 Das	21 Das
Control (C)	58.67 a	62.67 a	4.60 a	30.45 a	0.2745	0.2691
Drone (DR)	26.00 b	11.33 b	2.02 b	5.55 b	0.3186	0.2940
Spray machine (SM)	16.00 b	5.00 b	0.79 b	2.52 b	0.3201	0.3229
*p* value	0.0275	0.0462	0.0029	0.0019	0.3399	0.4647

## Data Availability

Data are contained within the article.

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
