# Peer review of "Remote Sensing Evaluation Drone Herbicide Application Effectiveness for Controlling Echinochloa spp. in Rice Crop in Valencia (Spain)"

_sensors, 2024, doi:10.3390/s24030804_

Round 1

Reviewer 1 Report

Comments and Suggestions for Authors

The authors do an excellent job of presenting their findings. Throughout the paper, the clarity can be improved by thinking about what the authors wish to convey to the reader and how to do so clearly. In the detailed comments, I indicate where I got completely lost. I hope this will lead to a more structured, coherent paper.

Abstract

- L 23-35: Can you elaborate on the specific objectives of evaluating the effectiveness of drone application of the herbicide chihalofop-butyl compared to sterilizing machines? What prompted this research, and what knowledge gap does it aim to fill? These lines are too lengthy, and I would like to see the summarize version.

- L 24-25: Delete it.

In comparing the effectiveness of the herbicide application with the drone and the bar, what specific metrics were used to assess efficacy? How do these metrics correlate with traditional measures such as the number of rice plants and weeds per area?

- At the end of this section, given that rice is a global staple, how transferable are the findings from this study to rice crops in other regions with different climates and environmental conditions?

Introduction:

- L 46-47: Use an update reference for this statement (citation from 2011 is too old).

- L 80: "The traditional way of controlling weeds...", in Spain or in the world?

 - Overall, try to decrease the numbers of paragraphs in this section (6 paragraphs would be great).

- L 129-136: Add research hypothesis here.

Materials and Methods:

- At the end of this section, I would like to see a new figure about the study flowchart.

Results:

- L 283-285: This is not belong to this section (move it to Materials and Methods).

- Figures 1-4: Add bars (standard error or standard deviation) to each column. Also, merge figures 1-4 in a single figure with different panels.

- Figure 6: Move it to supplementary file. 

Discussion:

At the end of this section, I would like to see research limitations, management implications, and future directions. Please add these as a separate paragraph.

Conclusion:

No comment.

Author Response

Response to Reviewer 1:

We thank the Reviewer 1 for his/her comments and suggestions.

Reviewer 1 question 1:

Abstract

L 23-35: Can you elaborate on the specific objectives of evaluating the effectiveness of drone application of the herbicide chihalofop-butyl compared to sterilizing machines? What prompted this research, and what knowledge gap does it aim to fill? These lines are too lengthy, and I would like to see the summarize version.

L 24-25: Delete it.

In comparing the effectiveness of the herbicide application with the drone and the bar, what specific metrics were used to assess efficacy? How do these metrics correlate with traditional measures such as the number of rice plants and weeds per area?

At the end of this section, given that rice is a global staple, how transferable are the findings from this study to rice crops in other regions with different climates and environmental conditions?

Following your recommendations, we have rewritten the abstract.

Reviewer 1 question 2:

Introduction

L 46-47: Use an update reference for this statement (citation from 2011 is too old).

We have replaced the initial reference with:

Bhandari, H. Global rice production, consumption and trade: trends and future directions. Proceedings of the Korean Society of Crop Science Conference. 2019. The Korean Society of Crop Science. pp. 5.

L 80: "The traditional way of controlling weeds...", in Spain or in the world?

            In the world. We have indicated it in the introduction:

“Globally, the traditional way of controlling weeds, pests and diseases in permanently waterlogged rice fields is complicated because the soil is not firm.”

 Overall, try to decrease the numbers of paragraphs in this section (6 paragraphs would be great).

We have reduced the number of paragraphs to 5. We have eliminated phrases from the introduction that we believe were not relevant.

L 129-136: Add research hypothesis here.

In the last paragraph of the introduction section we have included the starting hypothesis..

Reviewer 1 question 3:

Materials and Methods:

At the end of this section, I would like to see a new figure about the study flowchart.

We have included the study flow chart in the manuscript: Figure 1.

Reviewer 1 question 4:

Results:

L 283-285: This is not belong to this section (move it to Materials and Methods).

            We have moved L283-285 to Materials and Methods section.

Figures 1-4: Add bars (standard error or standard deviation) to each column. Also, merge figures 1-4 in a single figure with different panels.

We have combined Figures 1 through 4 into a single figure with different panels and added standard error bars to each column.

Figure 6: Move it to supplementary file. 

We have moved Figure 6 to supplementary file.

Reviewer 1 question 5:

Discussion:

At the end of this section, I would like to see research limitations, management implications, and future directions. Please add these as a separate paragraph.

We have added the following paragraph at the end of the Discussion section:

“The monitoring of herbicide applications by a drone using Sentinel 2 images could help to study and better understand the dynamics of weed growth in rice crop, for a better integrated weed management. Furthermore, they could be very beneficial for weed control, as they are more sustainable in economic and environmental terms, as shown by other studies [57, 61, 62, 63], especially in more protected natural environments such as the Albufera Natural Park. (Valencia, Spain). Several studies have shown that herbicides can be used to control invasive plants in managed wetlands [64].”

Reviewer 2 Report

Comments and Suggestions for Authors

Dear Authors I appreciate the manuscript that report an interesting field trial

The remote sensing part need to be strengthen, the word "change detection" do not figured in the manuscript, it would be nice to refer to the original principles of the remote sensing technology for the detection of the herbicide efficacy when analysing NDVI series.

there was a group did some research on rice I can suggest to have a look at this 

  •  
  • 10.1007/s13593-018-0548-9

Comments on the Quality of English Language

Language need a revision, but before the language, a remote sensing expert will improve the description of the methods using the terminology of the community of remote sensing sciences

Author Response

Response to Reviewer 2:

We thank the Reviewer 2 for his/her comments and suggestions.

Reviewer 2 question 1:

The remote sensing part need to be strengthen, the word "change detection" do not figured in the manuscript, it would be nice to refer to the original principles of the remote sensing technology for the detection of the herbicide efficacy when analysing NDVI series.

We have rewritten the abstract and in subsection 2.3. Monitoring using Sentinel 2 images, we have included the following paragraph:

One of the key technologies that has improved agriculture in recent years is remote sensing. Satellite sensors obtain vegetation reflectance data at different wavelengths. In addition to obtaining multispectral data, they are able to obtain time series of multitemporal reflectance data. Depending on the phenological state of the vegetation, the reflectance of the vegetation is high in the green wavelength region and highest in the NIR region, while it is low in the blue and red wavelength regions of the visible spectrum. Thus, by algebraically combining these bands, in addition to the vegetation indices, different vegetation characteristics can be obtained.”

Reviewer 2 question 2:

There was a group did some research on rice I can suggest to have a look at this 10.1007/s13593-018-0548-9

We have included a new reference in the text:

  1. Mascanzoni, E.; Perego, A.; Marchi, N.; Scarabel, L.; Panozzo, S.; Ferrero, A.; Acutis, M.; Sattin, M. Epidemiology and agronomic predictors of herbicide resistance in rice at a large scale.  Sustain. Dev., 2018, 38, 68.

Reviewer 3 Report

Comments and Suggestions for Authors

The paper is very well written and presents interesting information. But, I am offering the following recommendations for further improvement:

1.     Data should represent the standard error bars in all the figures. The axis should be labelled accurately.

2.     For Figure 5, the label on Y- axis is not readable. Full forms of the treatments should be provided.

3.     Line 580 should be rephrased.

4.     The conclusion should provide future directions of the study.

Comments on the Quality of English Language

Manuscript require English grammar check for improvement in sentence formation. 

Author Response

Response to Reviewer 3:

We thank the Reviewer 3 for his/her comments and suggestions.

Reviewer 3 question 1:

 “Data should represent the standard error bars in all the figures. The axis should be labelled accurately.”

We have added standard error bars to each column.

Reviewer 3 question 2:

“For Figure 5, the label on Y- axis is not readable. Full forms of the treatments should be provided.”

We have replaced the figure and now the label on Y-axis is readable

Reviewer 3 question 3:

“Line 580 should be rephrased.”

We have rewritten the first paragraph of the Conclusions section:

“This work validated the use of remote sensing to evaluate the effectiveness of the application of the herbicide cihalofop-butyl high in the early stages of rice growth (7 days after sowing). The results were compared when the herbicide was applied using a spraying machine and when it was applied using drones.”

Reviewer 3 question 4:

“The conclusion should provide future directions of the study.”

We have rewritten the last paragraph of the Conclusions section:

“The experiment reported the feasibility and high efficiency of remote sensing to evaluate the effectiveness of herbicide application in rice crop. Although the results obtained in this work are based on the rice crop, they could be applied to other crops depending on the phenological state of the crop and the active ingredient of the herbicide. These results could also be applied to other areas with different climatic and environmental conditions. “

Round 2

Reviewer 1 Report

Comments and Suggestions for Authors

I am happy with the revised version, congratulation!